# Effect of Physiological Concentrations of Vitamin C on the Inhibitation of Hydroxyl Radical Induced Light Emission from Fe^2+^-EGTA-H_2_O_2_ and Fe^3+^-EGTA-H_2_O_2_ Systems In Vitro

**DOI:** 10.3390/molecules26071993

**Published:** 2021-04-01

**Authors:** Michal Nowak, Wieslaw Tryniszewski, Agata Sarniak, Anna Wlodarczyk, Piotr J. Nowak, Dariusz Nowak

**Affiliations:** 1Radiation Protection, University Hospital No. 2, Medical University of Lodz, Zeromskiego 113, 90-549 Lodz, Poland; m.nowak@skwam.lodz.pl; 2Department of Radiological and Isotopic Diagnostics and Therapy, Medical University of Lodz, Zeromskiego 113, 90-549 Lodz, Poland; wieslaw.tryniszewski@umed.lodz.pl; 3Department of General Physiology, Medical University of Lodz, Mazowiecka 6/8, 92-215 Lodz, Poland; agata.sarniak@umed.lodz.pl; 4Department of Sleep Medicine and Metabolic Disorders, Medical University of Lodz, Mazowiecka 6/8, 92-215 Lodz, Poland; anna.wlodarczyk@umed.lodz.pl; 5Department of Nephrology, Hypertension, and Kidney Transplantation, Medical University of Lodz, Pomorska 251, 92-213 Lodz, Poland; piotr.nowak@umed.lodz.pl; 6Department of Clinical Physiology, Medical University of Lodz, Mazowiecka 6/8, 92-215 Lodz, Poland

**Keywords:** ascorbic acid, dehydroascorbic acid, chemiluminescence, Fenton system, antioxidant activity, pro-oxidant activity

## Abstract

Ascorbic acid (AA) has antioxidant properties. However, in the presence of Fe^2+^/Fe^3+^ ions and H_2_O_2_, it may behave as a pro-oxidant by accelerating and enhancing the formation of hydroxyl radicals (^•^OH). Therefore, in this study we evaluated the effect of AA at concentrations of 1 to 200 µmol/L on ^•^OH-induced light emission (at a pH of 7.4 and temperature of 37 °C) from 92.6 µmol/L Fe^2+^—185.2 µmol/L EGTA (ethylene glycol-bis (β-aminoethyl ether)-N,N,N′,N′-tetraacetic acid)—2.6 mmol/L H_2_O_2_, and 92.6 µmol/L Fe^3+^—185.2 µmol/L EGTA—2.6 mmol/L H_2_O_2_ systems. Dehydroascorbic acid (DHAA) at the same range of concentrations served as the reference compound. Light emission was measured with multitube luminometer (AutoLumat Plus LB 953) for 120 s after automatic injection of H_2_O_2_. AA at concentrations of 1 to 50 µmol/L and of 1 to 75 µmol/L completely inhibited light emission from Fe^2+^-EGTA-H_2_O_2_ and Fe^3+^-EGTA-H_2_O_2_, respectively. Concentrations of 100 and 200 µmol/L did not affect chemiluminescence of Fe^3+^-EGTA-H_2_O_2_ but tended to increase light emission from Fe^2+^-EGTA-H_2_O_2_. DHAA at concentrations of 1 to 100 µmol/L had no effect on chemiluminescence of both systems. These results indicate that AA at physiological concentrations exhibits strong antioxidant activity in the presence of chelated iron and H_2_O_2_.

## 1. Introduction 

Reactive oxygen species (ROS) are involved in numerous physiological and pathological processes in the human body [1,2]. ROS include various radicals and oxidants and among them the most reactive species are hydroxyl radicals (^•^OH) [1]. Iron (Fe^2+^) dependent reduction of H_2_O_2_ (Fenton chemistry) is the major source of ^•^OH radicals [1,3]. However, other transition metals such as copper, manganese, vanadium [4,5], as well as ionizing radiation [6,7] and peroxynitrite [8,9] could also contribute to generation of ^•^OH radicals in vivo. The reaction of Fe^2+^ with H_2_O_2_ initiates numerous radical and non-radical processes, leading to the formation of ^•^OH radicals, Fe^3+^, superoxide radicals (O_2•_^−^), and singlet oxygen [10,11,12]. O_2•_^−^ can reduce Fe^3+^ to Fe^2+^ and this in turn accelerates the formation of ^•^OH radicals [10,11]. Moreover, H_2_O_2_ can directly react with Fe^3+^ with subsequent creation of Fe^2+^, hydroperoxyl radical (HO_2_^•^), and H^+^ [11]. Thus, one may say that Fe^2+^ ions are regenerated and catalyze the reduction of H_2_O_2_ into ^•^OH radicals. It is well known that chemical compounds which can effectively reduce Fe^3+^ to Fe^2+^ added to Fenton’s reagent (aqueous solution containing Fe^2+^ and H_2_O_2_) strongly enhanced ^•^OH radicals generation. This was noted for hydroxylamine, ascorbic acid (AA), cysteine, 3-hydroxyanthranilic acid [13,14], and plant phenolics such as gallic acid, phloroglucinol, 3,4-dihydroxyphenylacetic acid, and phloretin [15]. Moreover, AA can undergo autooxidation, generating additional amounts of H_2_O_2_ [16,17]. Application of Fe^2+^-regenerating compounds appears to be a promising approach for Fenton chemistry in order to develop the effective methods of degradation of hazardous pollutants [13,14]. However, in the case of ascorbic acid (AA), this activity seems to be a double-edged sword. This is because H_2_O_2_ and free Fe^2+^ and Fe^3+^ ions (labile plasma iron) when present in circulating blood [18,19,20] AA may promote the generation of ^•^OH radicals and induction of the peroxidative damage to a variety of biomolecules [20]. It should be pointed out that combinations of iron ions with AA and/or H_2_O_2_ were widely used for the induction of DNA damage in in vitro studies [21,22,23,24]. On the other hand, this mechanism was proposed as one of those responsible for the killing of cancer cells by high AA concentrations [16,17,25]. Recently, we developed a system composed of Fe^2+^, EGTA (ethylene glycol-bis (β-aminoethyl ether)-N,N,N′,N′-tetraacetic acid) and H_2_O_2_, which is a source of ^•^OH radicals-induced ultra-weak photon emission (UPE) [26]. ^•^OH radicals generated in the reaction of Fe^2+^ with H_2_O_2_ can attack and cleave the ether bond in an EGTA backbone structure. This leads to the formation of products containing triplet excited carbonyl groups and photons emissions [26]. Some studies indicate that the effect of AA on ^•^OH generation by Fenton’s reagent may depend on its concentration and time of addition from the onset of reaction of Fe^2+^ with H_2_O_2_ [13]. Furthermore, AA itself can react with ^•^OH radicals [27,28] while making the chemistry aspects of this process more complicated. Therefore, in this study we examined the effect of AA at a wide range of concentrations (including those present in human plasma) on the UPE of an Fe^2+^-EGTA-H_2_O_2_ system, as well as the UPE of medium containing Fe^3+^-EGTA-H_2_O_2_. The activity of AA was compared with that revealed by dehydroascorbic acid (DHAA), a relatively stable product of AA oxidation [13].

## 2. Results 

The light emission (UPE-ultra weak photon emission) from 92.6 µmol/L Fe^2+^—185.2 µmol/L EGTA—2.6 mmol/L H_2_O_2_ system was 2306 ± 910 (2052; 71) RLU (*n* = 11). The UPE from incomplete control systems Fe^2+^-H_2_O_2_ and Fe^2+^-EGTA was significantly lower (*p* < 0.05, *n* = 11) and reached 1022 ± 295 (958; 339) RLU and 675 ± 111 (678; 137) RLU, respectively. These results are in agreement with those previously described [26], which also showed no light emissions from EGTA-H_2_O_2_ and H_2_O_2_ alone in comparison to the medium alone.

### 2.1. Effect of Ascorbic Acid and Dehydroascorbic Acid on Light Emission from Fe^2+^-EGTA-H_2_O_2_

The addition of AA (final concentrations of 1, 5, 10, 25, and 50 µmol/L) to the Fe^2+^-EGTA-H_2_O_2_ system completely abolished light emission (Figure 1A). The percent inhibition of UPE for concentrations of AA of 1, 5, 10, 25, and 50 µmol/L reached 102.3 ± 5.9 (99.7; 4.8), 101.7 ± 4.6 (101.7; 5.6), 100.2 ± 5.1 (100.0; 5.3), 100.7 ± 3.1 (99.3; 2.8), and 96.6 ± 2.4 (96.9; 1.0) (*p* < 0.05), respectively. The inhibition percentage slightly higher than 100 was most probably due to fluctuation of the baseline signal from medium alone. Higher concentrations of AA did not suppress the UPE of Fe^2+^-EGTA-H_2_O_2_ (Figure 1A). Even at concentrations of 100 µmol/L and 200 µmol/L, a moderate tendency (but not significant) of the enhancement of light emission was noted (Figure 1A). Control experiments showed no effect of AA (final concentrations of 1 and 200 µmol/L) on the UPE of incomplete systems (Fe^2+^-H_2_O_2_ and Fe^2+^-EGTA-H_2_O) and medium alone (Figure 1B). In these cases, the light emissions did not differ from the signal noted for medium alone. 

All tested concentrations of DHAA had no effect on the UPE of the Fe^2+^-EGTA-H_2_O_2_ system (Figure 2A), while the DHAA at concentrations of 1 and 200 µmol/L did not alter the light emission from the control systems, Fe^2+^-H_2_O_2_, Fe^2+^-EGTA-H_2_O, or the medium alone (Figure 2B).

### 2.2. Effect of Ascorbic Acid and Dehydroascorbic Acid on Light Emission from Fe^3+^-EGTA-H_2_O_2_


The Fe^3+^-EGTA-H_2_O_2_ system was a weaker light emitter than the Fe^2+^-EGTA-H_2_O_2_ system [1531 ± 292, (1493; 443) RLU, *n* = 8 vs. 2306 ± 910 (2052; 71) RLU, *n* = 11, *p* < 0.05]. Addition of AA to the Fe^3+^-EGTA-H_2_O_2_ system to final concentrations of 1, 5, 10, 25, 50, and 75 µmol/L completely abolished the UPE (Figure 3A) and the percent inhibition of the UPE was 103.1 ± 5.8 (105.1; 3.9), 92.5 ± 22.0 (102.3; 13.2), 102.4 ± 1.7 (102.8; 2.2), 106.5 ± 2.7 (105.3; 1.6), 106.3 ± 3.5 (106.5; 3.2), and 100.5 ± 3.6 (101.4; 4.0), respectively. AA at concentrations of 100 and 200 µmol/L did not significantly change the UPE of the Fe^3+^-EGTA-H_2_O_2_ system (*p* > 0.05, Figure 3A). AA at concentrations of 1 and 200 µmol/L decreased the light emission from the Fe^3+^-H_2_O_2_ (*p* < 0.05) but had no effect on Fe^3+^-EGTA-H_2_O and medium alone (Figure 3B).

DHAA at concentrations of 1, 5, 10, 25, and 50 µmol/L had no effect on the UPE of the Fe^3+^-EGTA-H_2_O_2_ system (Figure 4A). For the concentrations of 75 and 100 µmol/L, a slight tendency to increase the light emission was noted, while at the DHAA concentration of 200 µmol/L, a tremendous increase in mean UPE (almost 70-times) was found (Figure 4A). DHAA at concentrations of 1 and 200 µmol/L also increased the light emission from Fe^3+^-H_2_O_2_ (*p* < 0.05) but had no effect on other controls (Fe^3+^-EGTA or medium alone) (Figure 4B). 

## 3. Discussion 

Numerous reactions can occur simultaneously in an Fe^2+^-EGTA-H_2_O_2_ system [12,13,14,29,30]. Those leading to the generation of ^•^OH radicals, superoxide radicals (O_2•_^−^), hydroperoxyl radicals (HO_2_^•^), singlet oxygen (O_2_(^1^∆g)), and the reduction of Fe^3+^ to Fe^2+^ are presented below:Fe^2+^-EGTA + H_2_O_2_ → Fe^3+^-EGTA + OH^−^ + ^•^OH (formation of hydroxyl radicals) (1)
Fe^3+^-EGTA + H_2_O_2_ → Fe^3+^OOH^−^-EGTA + H^+^(2)
Fe^3+^OOH^−^-EGTA + H_2_O_2_ → FeO^2+^-EGTA + HO_2_^•^ + H_2_O (3)
FeO^2+^-EGTA + H_2_O_2_ → Fe^3+^-EGTA + HO_2_^•^ + OH^−^(4)
HO_2_^•^ → H^+^ + O_2•_^−^ (formation of superoxide radicals)(5)
Fe^3+^-EGTA + O_2•_^−^ → Fe^2+^-EGTA + O_2_(6)
Fe^3+^-EGTA + H_2_O_2_ → Fe^2+^-EGTA + HO_2_^•^(7)
Fe^3+^-EGTA + HO_2_^•^ → Fe^2+^-EGTA + O_2_ + H^+^.(8)

Hydroperoxyl radicals (HO_2_^•^) are generated in reactions (3), (4), and (7).

Reduced iron formed in reactions (6), (7), and (8) can enter reaction 1 to enhance the generation of hydroxyl radicals
O_2•_^−^ + ^•^OH + H^+^ → H_2_O_2_ + O_2_(^1^∆g) formation of singlet oxygen (9)
2O_2•_^−^ + 2H^+^ → H_2_O_2_ + O_2_(^1^∆g) formation of singlet oxygen.(10)

Hydroxyl radicals generated in an Fe^2+^-EGTA-H_2_O_2_ system can cleave one of the ether bonds in the backbone structure of EGTA, leading to formation of products with a triplet excited carbonyl group responsible for light emission [26]. O_2•_^−^ radicals can reduce Fe^3+^ into Fe^2+^ ions, which again react with H_2_O_2_ and generate ^•^OH radicals. Therefore, the UPE of the Fe^2+^-EGTA-H_2_O_2_ system was strongly inhibited by scavengers of ^•^OH radicals (dimethylsulfoxide and mannitol) and partially by superoxide dismutase, which very rapidly catalyzes the dismutation of O_2•_^−^ radicals into O_2_ and H_2_O_2_ [26]. The rates of reactions (7) and (8) are much slower than that of the reaction (1) [13,14]. Therefore, the UPE of Fe^2+^-EGTA-H_2_O_2_ was higher than the light emissions from Fe^3+^-EGTA-H_2_O_2_. In both Fe^2+^-EGTA-H_2_O_2_ and Fe^3+^-EGTA-H_2_O_2_ systems, the iron concentration was 28-times lower than the concentration of H_2_O_2_. Therefore, one may expect that the addition of reducing agent to both systems would increase the light emissions via the regeneration of Fe^2+^ ions and enhanced ^•^OH radicals formation. 

### 3.1. Effect of Ascorbic Acid and Dehydroascorbic Acid on the Light Emission from Fe^2+^-EGTA-H_2_O_2_

AA is a powerful reducer of Fe^3+^ ions [13,14]. Its ability to reduce Fe^3+^ ions into Fe^2+^ ions is stronger than that of uric acid, bilirubin, Trolox, and numerous plant phenolics such as ferulic acid, catechin, gallic acid, and quercetin [31,32]. Therefore, we expected that the addition of ascorbic acid to an Fe^2+^-EGTA-H_2_O_2_ system would enhance its UPE. Surprisingly, AA at concentrations of 1 to 50 µmol/L completely abolished the light emissions. Only higher AA concentrations of 100 and 200 µmol/L tended to increase the UPE of Fe^2+^-EGTA-H_2_O_2_ but this effect was not significant. AA can react with various reactive oxygen species such as H_2_O_2_, O_2•_^−^, O_2_(^1^∆g), and especially ^•^OH radicals [33,34,35]. In our previous experiments, sodium azide as a scavenger of O_2_(^1^∆g) did not suppress the light emission from Fe^2+^-EGTA-H_2_O_2_ and the contribution of O2_•_^−^ to this phenomenon was relatively low [26]. Moreover, the concentration of H_2_O_2_ was many times higher than that of AA, Fe^2+^, and EGTA in the Fe^2+^-EGTA-H_2_O_2_ system. Therefore, the plausible reaction of AA with H_2_O_2_ and O_2•_^−^ seems to not be responsible for the quenching of the UPE. Thus, the reaction of AA with ^•^OH radicals could have a crucial effect on the UPE of the Fe^2+^-EGTA-H_2_O_2_ system. The reaction of AA with ^•^OH radical leads to the formation of ascorbate radical and H_2_O. At a physiological pH, the reaction of disproportionation of two molecules of ascorbate radicals is thermodynamically favored. This is a complex process and involves dimerization of an ascorbate radical, internal electron transfer, and hydrolysis of temporal dimer, and results in the formation of one molecule of DHAA and AA [35,36] which can again react with an ^•^OH radical. The rate of the reaction of AA with Fe^3+^, which promotes ^•^OH radicals generation, was affected by the pH of the solution and at the pH higher than 6, the rate was slow [37]. Therefore, under conditions of our experiments (pH = 7.4), the reaction of AA with ^•^OH radicals dominates and protects molecules of EGTA from oxidative attack and generation of end-products, with triplet excited carbonyl groups responsible for light emission. These may explain the strong inhibitory effect of AA at concentrations of 1 to 50 µmol/L on the UPE of Fe^2+^-EGTA-H_2_O_2_. However, AA at higher concentrations of 75 to 200 µmol/L did not inhibit the UPE of Fe^2+^-EGTA-H_2_O_2_. This suggests that under those conditions, there is a relative balance between ^•^OH radicals generation and their scavenging by AA and thus the activity of ^•^OH radicals is similar in Fe^2+^-EGTA-H_2_O_2_ with and without high concentrations of AA. AA has chelating activity and was reported to form complexes with Fe^2+^ and Fe^3+^ ions [33,38]. Moreover, the formation of AA-Fe^3+^ complexes is necessary for AA- induced reduction of Fe^3+^ to Fe^2+^ [37,38]. EGTA is a chelating agent which complexes Fe^2+^ and Fe^3+^ ions [39]. In experiments with concentrations of AA of 75 to 200 µmol/L (close to concentration of EGTA of 185.2 µmol/L), there is a substantial possibility of formation of AA-Fe^3+^ complexes. Moreover, it cannot be excluded that under these conditions, mixed chelate complexes of EGTA-AA-Fe^3+^ could be formed. This is supported by the description of Fe^3+^- deferiprone-AA complexes (deferiprone is an iron chelator indicated for the treatment of iron overload) in medium of pH = 7.4 in vitro [33]. Thus, at concentrations of AA of 75 to 200 µmol/L, considerable augmentation of Fe^2+^ ions regeneration can occur. Therefore, the intensities of two reactions: AA- induced scavenging of ^•^OH radicals and AA- induced Fe^2+^ regeneration, are comparable and these explain why higher AA concentrations did not inhibit the UPE of the Fe^2+^-EGTA-H_2_O_2_ system. On the other hand, low concentrations of AA (1 to 50 µmol/L) could not form sufficient amounts of redox active complexes with Fe^3+^ due to an excess of EGTA. These outcomes additionally explain the strong inhibitory effect of low AA concentrations on light emission from the Fe^2+^-EGTA-H_2_O_2_ system. Figure 5 summarizes the mechanism of inhibitory effect of AA (concentrations of 1 to 50 µmol/L) on the ^•^OH radicals-induced UPE of the Fe^2+^-EGTA-H_2_O_2_ system. Although DHAA can react with H_2_O_2_ and ^•^OH radicals [40], no effect of DHAA on UPE of Fe^2+^-EGTA-H_2_O_2_ was noted. DHAA is the product of two-electron oxidation of AA [36]. Therefore, it is a much weaker electron donor than AA and the involvement of DHAA in redox reactions after an addition to Fe^2+^-EGTA-H_2_O_2_ was many times lower than that in the case of AA. Hence, DHA did not alter the light emission from the Fe^2+^-EGTA-H_2_O_2_ system.

### 3.2. Effect of Ascorbic Acid and Dehydroascorbic Acid on the Light Emission from Fe^3+^-EGTA-H_2_O_2_

As was stated before, the Fe^3+^-EGTA-H_2_O_2_ system was a weaker light emitter than the Fe^2+^-EGTA-H_2_O_2_ one. ^•^OH radicals initiating the light emission from Fe^3+^-EGTA-H_2_O_2_ system are formed in the reaction (1), which occurs as a result of the reaction (7). AA at concentrations of 1 to 75 µmol/L inhibited the UPE of the Fe^3+^-EGTA-H_2_O_2_ system through direct scavenging of ^•^OH radicals. Due to a medium pH of 7.4 and excess of EGTA, these low concentrations of AA could not effectively reduce Fe^3+^ ions to Fe^2+^ ions, therefore the inhibition of light emission was complete. However, at higher concentrations of AA (100 and 200 µmol/L), the process of Fe^2+^ ions formation was enhanced and resulted in higher generation of ^•^OH radicals. Thus, AA at concentrations of 100 and 200 µmol/L did not alter the light emission from the Fe^3+^ EGTA-H_2_O_2_ system due to a dynamic balance between ^•^OH radicals scavenging and the promotion of ^•^OH radicals generation caused by this vitamin. Because DHAA is a weaker electron donor than AA [36], this compound at concentrations of 1 to 100 µmol/L had no significant effect on the UPE of the Fe^3+^-H_2_O_2_-EGTA system. However, the concentration of DHAA of 200 µmol/L tremendously (by about 70-times) increased photons emission from the Fe^3+^-H_2_O_2_-EGTA system. DHAA was reported to react with H_2_O_2_ through the formation of 4-O-oxalyl-threonate and 3-O-oxalyl-threonate as the main products, small amounts of cyclic oxalyl-threonate, 2-keto-L-xylonate, and threonic acid, and trace amounts of oxalic acid while oxidation of DHAA by ^•^OH radicals generated by Fenton’s reagent (Fe^2+^-EDTA-H_2_O_2_) produced mainly oxalic acid and both isomers of oxalyl threonate and small amounts of threonic acid [40]. Because DHAA at a concentration of 200 µmol/L had no effect on light emission from Fe^2+^-EGTA-H_2_O_2_ and Fe^2+^-H_2_O_2_ as well as Fe^3+^-EGTA-H_2_O_2_ generated substantially less ^•^OH radicals than Fe^2+^-EGTA-H_2_O_2_, one may conclude that reactions leading to formation of cyclic oxalyl-threonate and 2-keto-L-xylonate may be involved in very strong augmentation of the UPE of the Fe^3+^-H_2_O_2_-EGTA system. DHAA also augmented the light emission from Fe^3+^-H_2_O_2_ by about 2.5-times, having no effect on this process in medium containing Fe^3+^ and EGTA. This suggests that EGTA is not necessary for moderate augmentation of UPE by DHAA oxidized in the presence of Fe^3+^ and H_2_O_2_. On the other hand, the combination of EGTA or its derivatives formed after ^•^OH- induced oxidative attack with cyclic oxalyl-threonate or 2-keto-L-xylonate may result in chemical reactions which efficiently generate light and strongly augment the UPE of the Fe^3+^-H_2_O_2_-EGTA system. However, confirmation of these hypothetical mechanisms requires further studies. 

### 3.3. Relevance to Human Physiology 

It is believed that plasma concentrations of H_2_O_2_ range from 1 to 5 µmol/L in healthy subjects. However, in the course of certain diseases, the levels of H_2_O_2_ in plasma can increase up to 50 µmol/L [18]. The plasma concentration of iron complexed with low molecular weight compounds is about 1 µmol/L in healthy subjects while in patients with hemochromatosis, this can reach 10 µmol/L [41]. The median concentrations of AA and DHAA in plasma of healthy subjects were around 61.4 µmol/L and 2.3 µmol/L [42], however in critically ill patients (sepsis, major-organ failure, severe accidental injury), they decreased to 9.0 µmol/L and 1.4 µmol/L, respectively [42]. Therefore, the studied concentrations of AA and DHAA included the concentration ranges which can occur in healthy subjects and those with a strong inflammatory response. Because oxygen pressure in arterial blood ranges from 75 mmHg to 100 mmHg in healthy subjects [43] and O_2_ is involved in final reactions, leading to the formation of a product with triplet excited carbonyl groups [26], we did not use deaerated solutions in our experiments. In addition, AA was stable in undeaerated phosphate buffers of pH = 7.2 and 7.8 for at least 50 min [44]. Thus, unspecific decompositions of AA could not have had any influence on the results of our experiments.

The most important finding was that AA at concentrations of 5 to 50 µmol/L which can occur in human plasma suppressed the ^•^OH radicals-induced light emission from both systems: Fe^2+^-EGTA-H_2_O_2_ and Fe^3+^-EGTA-H_2_O_2_. Moreover, the concentration of AA of 75 µmol/L inhibited the UPE of the Fe^3+^-EGTA-H_2_O_2_ but had no significant effect on that of Fe^2+^-EGTA-H_2_O_2_.These suggest that under physiological conditions, the antioxidant activity (scavenging of ^•^OH radicals) of AA prevails over its plausible pro-oxidant activity related to the reduction of Fe^3+^ to Fe^2+^ ions. It should be pointed out that even higher concentrations of AA of 100 and 200 µmol/L did not significantly alter the UPE of Fe^2+^-EGTA-H_2_O_2_ and Fe^3+^-EGTA-H_2_O_2_ systems. However, circulating blood plasma containing H_2_O_2_ and iron complexed with low molecular weight chelating compounds is much more complex medium than our in vitro model. Recent clinical studies showing that intravenous administration of AA in a single dose of 750 mg or 7500 mg for six days did not increase oxidative stress markers (plasma concentrations of thiobarbituric acid reactive substances and urinary 8-oxoguanosine) in healthy subjects [45] support our observations.

Circulating blood has a pH of around 7.4 and a temperature of 37 °C. However, locally at the place of inflammation and also in certain solid tumors, the tissue environment could be acidic with a pH ranging from 5.7 to 7.0 [46]. This may predispose towards the reduction of Fe^3+^ to Fe^2+^ by AA and the enhanced generation of ^•^OH radicals. Therefore, pro-oxidant activity of ascorbate cannot be excluded under such circumstances. 

### 3.4. Limitations of the Study 

The UPE was measured with a luminometer equipped with a photon counter sensitive to photons emitted in the 380 nm–630 nm range. We proposed a mechanism of light emission by Fe^2+^-EGTA-H_2_O_2_ which involves an ^•^OH- induced cleavage of the ether bond in the backbone chain of EGTA molecule, its further degradation and formation of another radical, and triplet excited carbonyl groups [26]. Triplet excited carbonyl groups emit photons with a spectral range of 350 nm to 550 nm [47]. The human body spontaneously emits light, mostly within the wavelength range of 420 nm to 570 nm [48]. This suggests the occurrence of other sources of UPE in body fluids than triplet excited carbonyl groups. Therefore, a lack of spectral analysis of the UPE of Fe^2+^-EGTA-H_2_O_2_ and Fe^3+^-EGTA-H_2_O_2_ with and without studied compounds could be recognized as the limitation of our study. Unfortunately, there were no technical capabilities to use any cut-off filters for spectral analysis in AutoLumat Plus LB 953. Spectral analysis of the UPE would be especially helpful for an explanation of 200 µmol/L DHAA-induced enhancement of light emission form Fe^3+^-EGTA-H_2_O_2_. If the emission spectra of Fe^3+^-EGTA-H_2_O_2_ and Fe^3+^-EGTA-DHAA-H_2_O_2_ would be similar, one may conclude that this enhancement of the UPE is the consequence of increased formation of triplet excited carbonyl groups. However, from the physiological and clinical point of view, this is not important because concentrations of DHAA of 200 µmol/L could not occur in human plasma, even after intravenous administration of high doses of AA [45]. On the other hand, it cannot be excluded that strong light emission from Fe^3+^-EGTA-DHAA-H_2_O_2_ may be used for the determination of anti-oxidant properties of other compounds. Therefore, further experiments to elucidate the mechanism of DHAA-induced augmentation of the UPE of Fe^3+^-EGTA-H_2_O_2_ are worth conducting.

## 4. Materials and Methods

### 4.1. Reagents 

All chemicals were of analytical grade. Iron (II) sulfate heptahydrate (FeSO_4_ × 7H_2_O), iron (III) chloride hexahydrate (FeCl_3_ × 6H_2_O), sodium L-ascorbate (AA), L-dehydroascorbic acid (DHAA), and ethylene glycol-bis (β-aminoethyl ether)-N,N,N′,N′-tetraacetic acid (EGTA) were purchased from Sigma-Aldrich Chemicals (St. Louis, MO, USA.) H_2_O_2_ 30% solution (*w*/*w*) was from Chempur (Piekary Slaskie, Poland). Sterile phosphate buffered saline (PBS, pH 7.4, without Ca^2+^ and Mg^2+^) was obtained from Biomed (Lublin, Poland). Sterile deionized pyrogen-free water (freshly prepared, resistance >18 MW/cm, HPLC H_2_O Purification System, USF Elga, Buckinghamshire, UK) was used throughout the study. Working aqueous solutions of 5 mmol/L of FeSO_4_ and 5 mmol/L of FeCl_3_ were prepared before the assay. A working solution of 28 mmol/L of H_2_O_2_ was also prepared before the assay by dilution of 30% of H_2_O_2_ solution and the concentration was confirmed by the measurement of absorbance at 240 nm using a molar extinction coefficient of 43.6 mol^−1^cm^−1^ [49]. A stock solution of EGTA (100 mmol/L) was prepared in PBS with pH adjusted to 8.0 with 5 mol/L of NaOH and was stored at room temperature in the dark for no longer than 3 months. Ten mmol/L of EGTA working solution was obtained by appropriate dilution of EGTA stock solution with water before the assay. AA and DHAA solutions in PBS (7.2 mmol/L) and their 2-, 2.7-, 4-, 8-, 20-, 40-, and 200-times dilutions were prepared freshly before the assay.

### 4.2. System Generating Light and Measurement of Light Emission

For light generation, we used 92.6 µmol/L Fe^2+^—185.2 µmol/L EGTA—2.6 mmol/L H_2_O_2_ system, as previously described [26]. This system generates the UPE, which depends mainly on ^•^OH radicals. ^•^OH generated in the course of reaction of Fe^2+^ with H_2_O_2_ can oxidatively attack and cleave ether bonds in EGTA molecule, which leads to formation of triplet excited carbonyl groups and light emission [26]. Briefly, 20 µL of 10 mmol/L EGTA solution was added to the tube (Lumi Vial Tube, 5 mL, 12 × 75 mm, Berthold Technologies, Bad Wildbad, Germany) containing 940 µL of PBS. Afterwards, 20 µL of 5 mmol/L solution of FeSO_4_ was added and after gentle mixing, the tube was placed in the luminometer chain and incubated for 10 min in the dark at 37 °C. Then, 100 µL of 28 mmol H_2_O_2_ solution was added by an automatic dispenser and the total light emission (expressed in RLU—relative light units) was measured for 120 s with a multitube luminometer (AutoLumat Plus LB 953, Berthold, Germany) equipped with a Peltier-cooled photon counter (spectral range from 380 to 630 nm) to ensure high sensitivity and low and stable background noise signals. 

### 4.3. Effect of Ascorbic Acid and Dehydroascorbic Acid on Light Emission from Fe^2+^-EGTA-H_2_O_2_ System

In order to determine the effect of AA on the UPE of 92.6 µmol/L Fe^2+^—185.2 µmol/L EGTA—2.6 mmol/L H_2_O_2_ system, 30 µL of working solution of AA in PBS or its appropriate dilutions were added to the luminometer tube containing EGTA and FeSO_4_ in PBS and incubated for 10 min at 37 °C in the dark and then 100 µL of H_2_O_2_ solution was injected and the total light emission was measured for 2 min. The final concentrations of AA in the reaction mixture were 1, 5, 10, 25, 50, 75, 100, and 200 µmol/L, respectively. Controls included: Fe^2+^-EGTA-H_2_O_2_ in PBS without AA, incomplete system Fe^2+^-H_2_O_2_ with and without AA, Fe^2+^-EGTA with and without AA, AA alone in PBS, and medium alone. The final concentrations of AA in the controls were 1 and 200 µmol/L. The same procedures were executed when the effect of DHAA on the UPE of 92.6 µmol/L Fe^2+^—185.2 µmol/L EGTA—2.6 mmol/L H_2_O_2_ was studied. The design of these experiments is shown in Table 1. In each series of experiments (repeated at least 4 times), eight concentrations of AA or DHAA were tested. The inhibitory effect of AA or DHAA on the light emission was expressed as a percent inhibition (%I) calculated according to the formula: %I = [(A − B)/(A − C)] × 100% where A, B, and C are the total light emission from Fe^2+^-EGTA-H_2_O_2_, Fe^2+^-EGTA-studied compound (AA or DHAA)-H_2_O_2_, and medium (H_2_O injected into PBS), respectively. In the case of augmentation of the UPE, the percent enhancement (%E) was calculated as follows: %E = [(B − A)/(A − C)] × 100%. In additional experiments, the effect of AA and DHAA on light emission from 92.6 µmol/L Fe^3+^—185.2 µmol/L EGTA—2.6 mmol/L H_2_O_2_ was examined. The design of these experiments was the same as in Table 1, except for the addition of 20 µL of working solution of FeCl_3_ instead of 20 µL of working solution of Fe_2_SO_4_.

### 4.4. Statistical Analysis 

Results (total light emission, % inhibition or % enhancement of light emission) were expressed as mean (standard deviation) and median and interquartile range (IQR). The comparisons between the UPE of the Fe^2+^-EGTA-H_2_O_2_ system and the light emission from corresponding samples of a modified system (e.g., an incomplete system, system with the addition of AA or DHAA, Fe^3+^-EGTA-H_2_O_2_ with and without addition of AA or DHAA, and medium alone) were analyzed with the independent-samples (unpaired) *t*-test or Mann-Whitney U test depending on the data distribution, which was tested with the Kolmogorov–Smirnov–Liliefors test. The Brown–Forsythe test for analysis of the equality of the group variances was used prior to the application of the unpaired *t*-test and if variances were unequal, then the Welch’s *t*-test was used instead of the standard *t*-test. The comparisons of % inhibition or % enhancement of light emission caused by AA and DHAA were analyzed in the same way. A *p*-value < 0.05 was considered significant.

## 5. Conclusions 

Ascorbic acid within the concentration range of 1 to 50 µmol/L very effectively inhibited ^•^OH-induced light emission from Fe^2+^-EGTA-H_2_O_2_ and Fe^3+^-EGTA-H_2_O_2_ systems in vitro. Higher concentrations of 75 to 200 µmol/L did not significantly enhance the UPE of both modified Fenton systems. Because studied concentrations of AA involved those present in human plasma, one may conclude that AA can act as an antioxidant in the presence of iron complexed with low molecular weight compounds in circulating blood. Dehydroascorbic acid within the range of physiological concentrations of 1 to 5 µmol/L had no effect on the intensity of ^•^OH- induced reaction, resulting in the light emission. Although these results were obtained from in vitro experiments, they strongly suggest the low risk of pro-oxidant activity of AA in healthy subjects. 

## Figures and Tables

**Figure 1 molecules-26-01993-f001:**
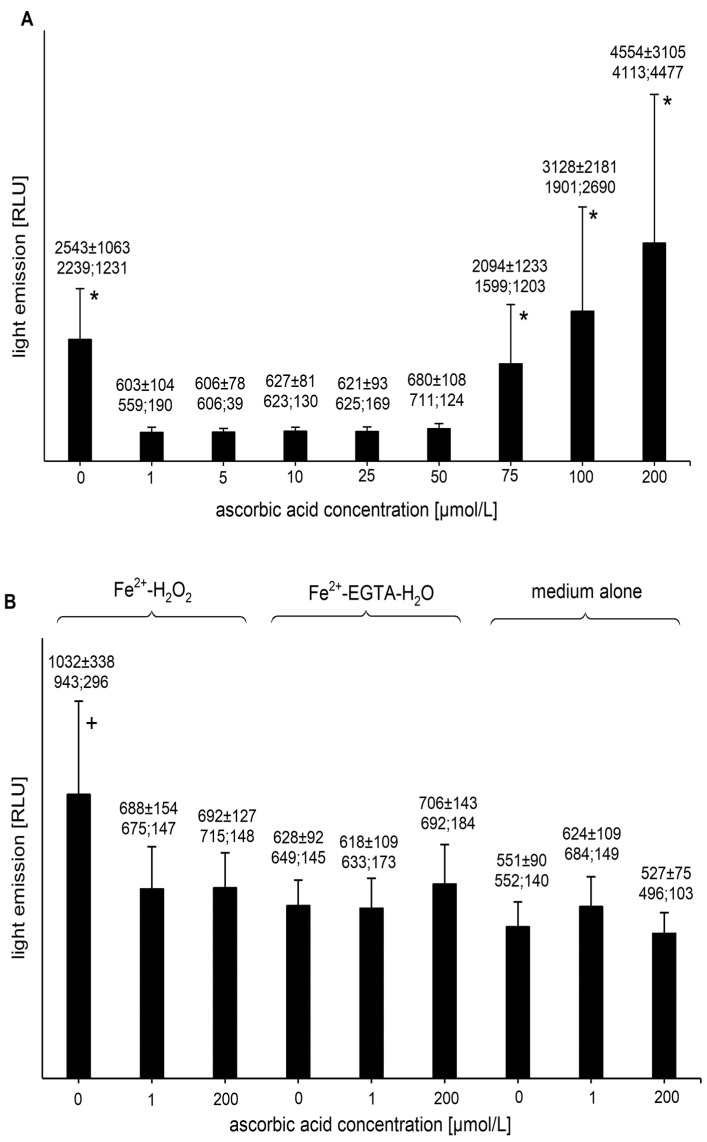
(**A**)—Effect of ascorbic acid on light emission from 92.6 µmol/L Fe^2+^—185.2 µmol/L EGTA—2.6 mmol/L H_2_O_2_ system (**B**)—Effect of ascorbic acid on light emission from control systems with 92.6 µmol/L Fe^2+^—2.6 mmol/L H_2_O_2_, 92.6 µmol/L Fe^2+^—185.2 µmol/L EGTA-H_2_O, and medium alone with H_2_O. Total light emission was measured for 2 min just after automatic injection of 100 µL of H_2_O_2_ solution or distilled water. The final sample volume was 1080 µL. Results obtained from seven series of experiments are expressed as mean and standard deviation and (median; interquartile range). * vs. ascorbic acid concentrations of 1, 5, 10, 25, and 50 µmol/L, *p* < 0.05. † vs. Fe^2+^-H_2_O_2_ with the addition of ascorbic acid at concentrations of 1 and 200 µmol/L, Fe^2+^-EGTA and medium alone with or without addition of ascorbic acid at concentration of 1 and 200 µmol/L, *p* < 0.05. EGTA–ethylene glycol-bis (β-amino ethyl ether)-N,N,N′,N′-tetra acetic acid, RLU—relative light units.

**Figure 2 molecules-26-01993-f002:**
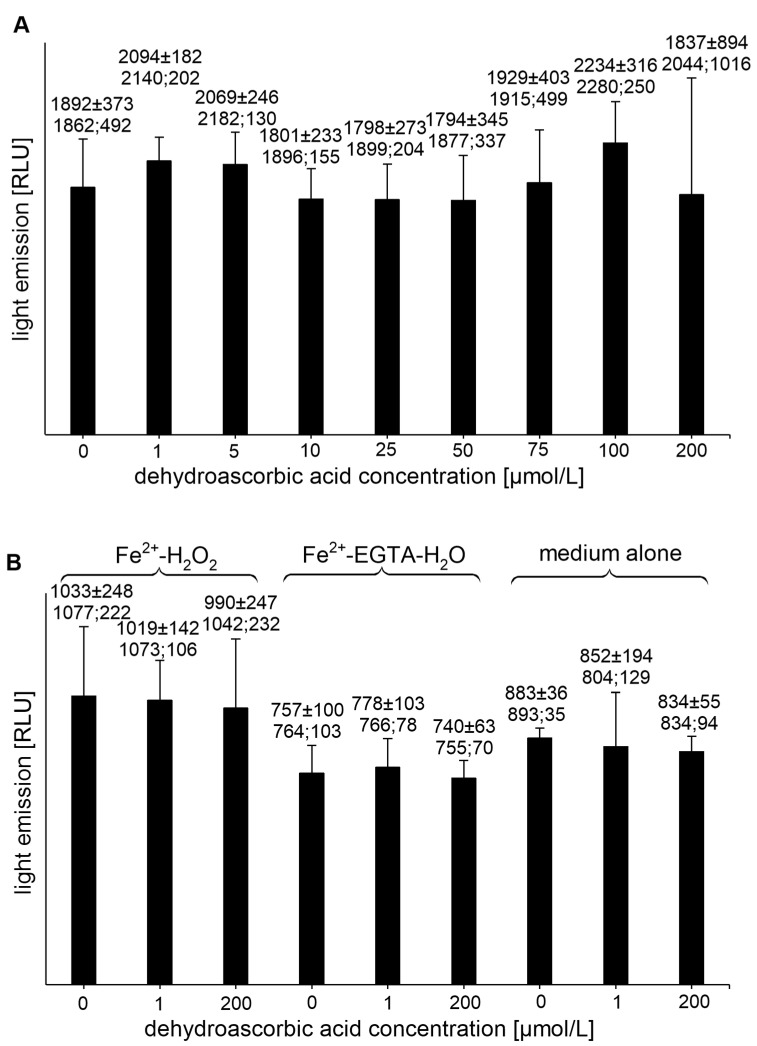
(**A**)—Effect of dehydroascorbic acid on light emission from 92.6 µmol/L Fe^2+^—185.2 µmol/L EGTA—2.6 mmol/L H_2_O_2_ system (**B**)—Effect of dehydroascorbic acid on light emission from control systems with 92.6 µmol/L Fe^2+^—2.6 mmol/L H_2_O_2_, 92.6 µmol/L Fe^2+^—185.2 µmol/L EGTA-H_2_O, and medium alone with H_2_O. Total light emission was measured for 2 min just after automatic injection of 100 µL of H_2_O_2_ solution or distilled water. Final sample volume: 1080 µL. Results obtained from four series of experiments expressed as mean and standard deviation and (median; interquartile range). EGTA—ethylene glycol-bis (β-amino ethyl ether)-N,N,N′,N′-tetra acetic acid, RLU—relative light units.

**Figure 3 molecules-26-01993-f003:**
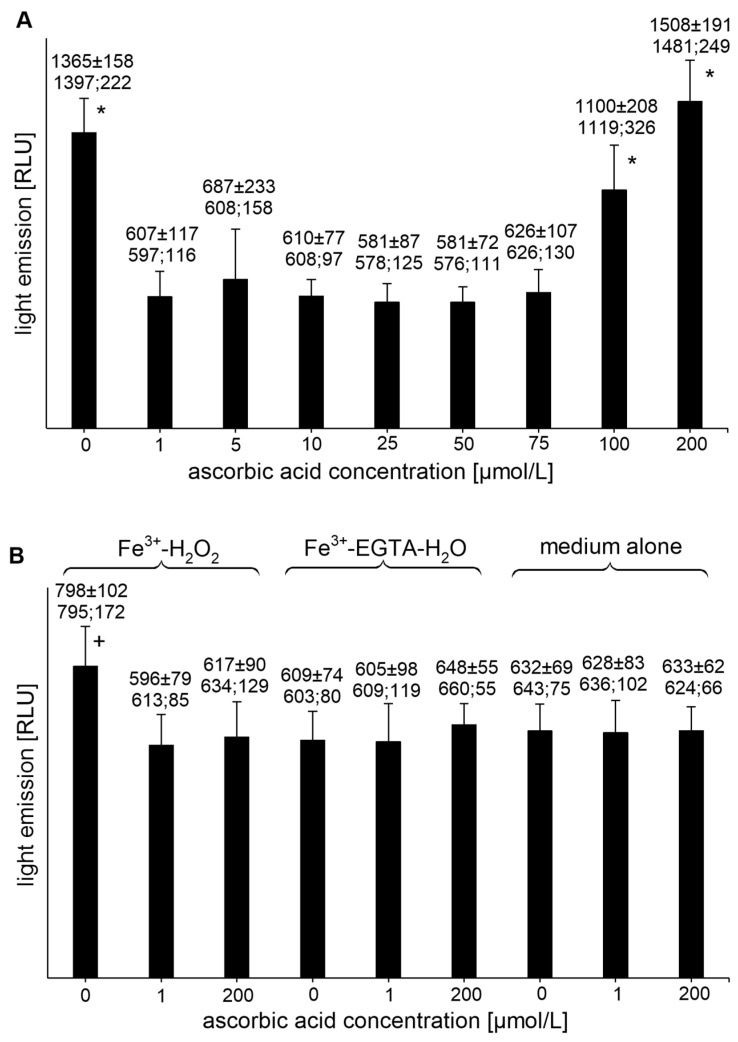
(**A**)—Effect of ascorbic acid on light emission from 92.6 µmol/L Fe^3+^—185.2 µmol/L EGTA—2.6 mmol/L H_2_O_2_ system (**B**)—Effect of ascorbic acid on light emission from control systems with 92.6 µmol/L Fe^3+^—2.6 mmol/L H_2_O_2_, 92.6 µmol/L Fe^3+^—185.2 µmol/L EGTA-H_2_O, and medium alone with H_2_O. Total light emission was measured for 2 min just after automatic injection of 100 µL of H_2_O_2_ solution or distilled water. Final sample volume: 1080 µL. Results obtained from four series of experiments expressed as mean and standard deviation and (median; interquartile range). * vs. ascorbic acid concentrations of 1, 5, 10, 25, 50, and 75 µmol/L, *p* < 0.05. † vs. Fe^3+^-H_2_O_2_ with addition of ascorbic acid at concentrations of 1 and 200 µmol/L, Fe^3+^-EGTA, and medium alone with or without the addition of ascorbic acid at concentrations of 1 and 200 µmol/L, *p* < 0.05. EGTA—ethylene glycol-bis (β-amino ethyl ether)-N,N,N′,N′-tetra acetic acid, RLU—relative light units.

**Figure 4 molecules-26-01993-f004:**
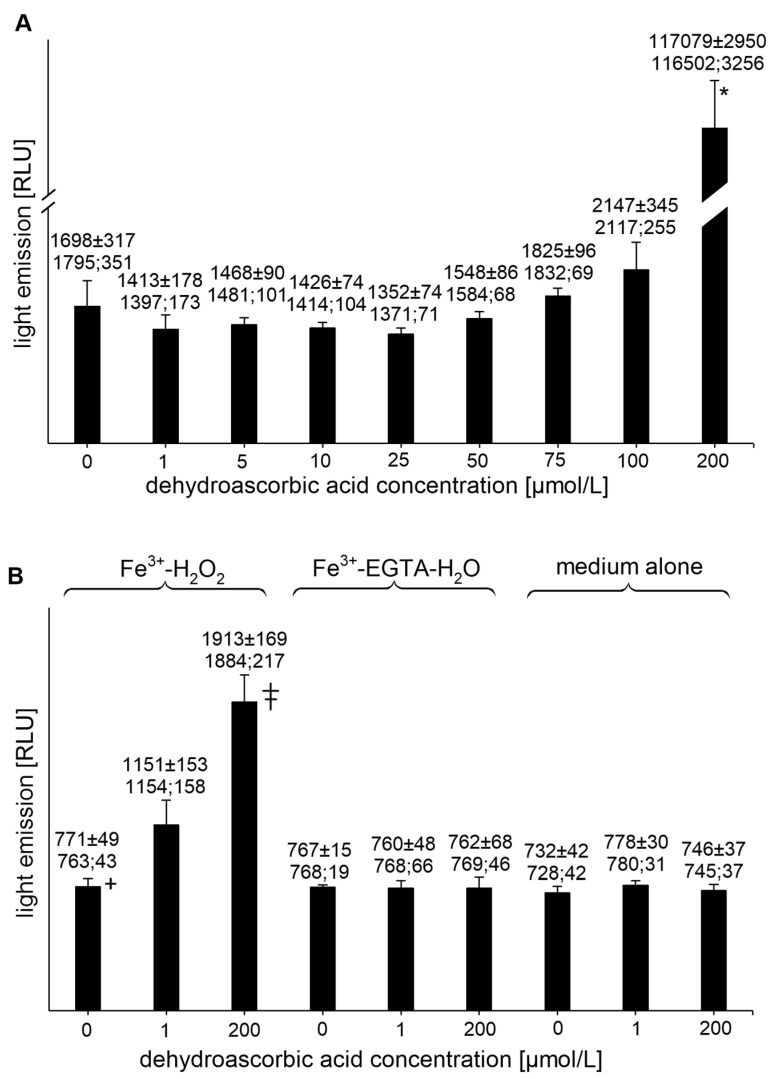
(**A**)—Effect of dehydroascorbic acid on light emission from 92.6 µmol/L Fe^3+^—185.2 µmol/L EGTA—2.6 mmol/L H_2_O_2_ system (**B**)—Effect of dehydroascorbic acid on light emission from control systems with 92.6 µmol/L Fe^3+^—2.6 mmol/L H_2_O_2_, 92.6 µmol/L Fe^3+^—185.2 µmol/L EGTA-H_2_O, and medium alone with H_2_O. Total light emission was measured for 2 min just after automatic injection of 100 µL of H_2_O_2_ solution or distilled water. Final sample volume: 1080 µL. Results obtained from four series of experiments expressed as mean and standard deviation and (median; interquartile range). * vs. dehydroascorbic acid concentrations of 0, 1, 5, 10, 25, 50, 75, and 100 µmol/L, *p* < 0.05. † vs. Fe^3+^—H_2_O_2_ with the addition of dehydroascorbic acid at concentrations of 1 and 200 µmol/L. ‡ vs. dehydroascorbic acid concentration of 1 µmol/L, *p* < 0.05. EGTA—ethylene glycol-bis (β-amino ethyl ether)-N,N,N′,N′-tetra acetic acid, RLU—relative light units.

**Figure 5 molecules-26-01993-f005:**
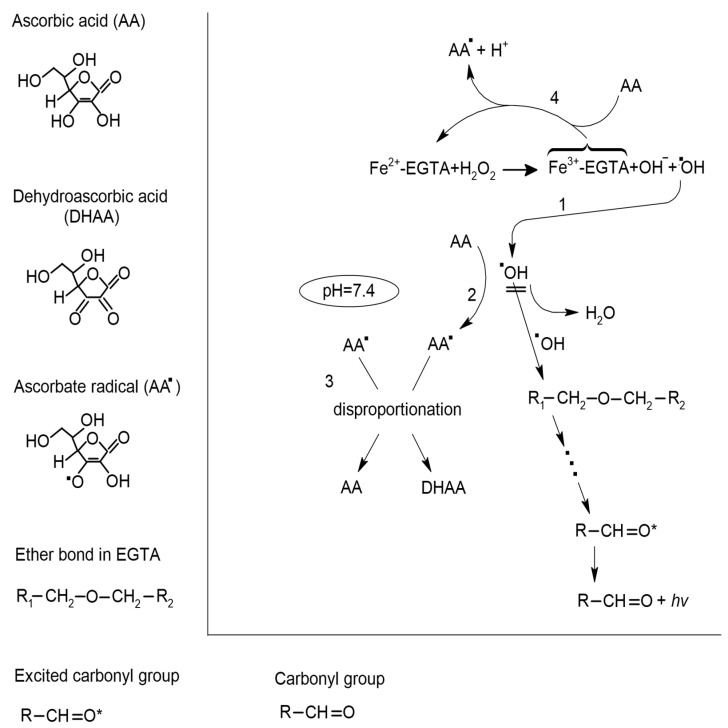
Postulated mechanism of inhibitory effect of ascorbic acid (AA) on the ^•^OH radical-induced ultraweak photon emission (UPE) of the Fe^2+^-EGTA-H_2_O_2_ system in medium of pH = 7.4. ^•^OH radicals generated in the reaction of Fe^2+^ with H_2_O_2_ (1) attack one of the two ether bonds in the backbone structure of EGTA, leading to its cleavage and formation of other radicals that results in the creation of one product with a triplet excited carbonyl group (R-CH = O *). Electronic transitions from the triplet excited state to the ground state are accompanied by the photon emission (λν). AA can effectively react with ^•^OH radicals (2) through the formation of an ascorbate radical (AA^•^). This protects the ether bonds of EGTA from oxidative attack and completely inhibits light emission from the Fe^2+^-EGTA-H_2_O_2_ system. Two molecules of AA^•^ undergo disproportionation reaction (3) with the formation of one molecule of dehydroascorbic acid (DHAA) and one of AA, which again can react with ^•^OH radicals. AA can reduce Fe^3+^ to Fe^2+^ (4) and therefore, enhances ^•^OH radicals generation. However, under conditions of pH = 7.4 and excess of EGTA as chelating agent, this process has low intensity. This pathway of Fe^2+^ regeneration is enhanced for higher concentrations of AA (75 to 200 µmol/L) and therefore, they do not inhibit the UPE of the Fe^2+^-EGTA-H_2_O_2_ system. For more details, please refer to [26,35,36].

**Table 1 molecules-26-01993-t001:** Design of experiments on the effect of ascorbic acid and dehydroascorbic acid on light emissions from the Fe^2+^-EGTA-H_2_O_2_ system.

No.	Sample	Volumes of Working Solutions Added to Luminometer Tube [µL]						
		A	B	C	D	E	F	G
		PBS	EGTA	FeSO_4_	AA	DHAA	H_2_O_2_	H_2_O
1	Complete system	940	20	20	-	-	100	-
2	Complete system + AA	910	20	20	30	-	100	-
3	Complete system + DHAA	910	20	20	-	30	100	-
4	Incomplete system	960	-	20	-	-	100	-
5 *	Incomplete system + AA	930	-	20	30	-	100	
6 *	Incomplete system + DHAA	930	-	20	-	30	100	-
Additional controls								
7	Fe^2+^-EGTA without H_2_O_2_	940	20	20	-	-	-	100
8 *	Fe^2+^-EGTA without H_2_O_2_ +AA	910	20	20	30	-	-	100
9 *	Fe^2+^-EGTA without H_2_O_2_ +DHAA	910	20	20	-	30	-	100
10 *	AA alone	950	-	-	30	-	-	100
11 *	DHAA alone	950	-	-	-	30	-	100
12	Medium alone	980	-	-	-	-	-	100

Working solutions were added to the luminometer tube in alphabetical order: A—sterile phosphate buffered saline (PBS) (pH = 7.4) without divalent cations; B—10 mmol/L aqueous solution of EGTA: C—5 mmol/L aqueous solution of FeSO_4_; D—solution of AA in PBS (final concentration of AA in the reaction mixture ranged from 1 to 200 µmol/L, eight concentrations); E—solution of DHAA in PBS (final concentration of DHAA in the reaction mixture ranged from 1 to 200 µmol/L). Then after gentle mixing the tube was placed into luminometer chain, incubated for 10 min at 37 °C, and then 28 mmol/L H_2_O_2_ (F) or water (G) was automatically injected by dispenser and the total light emission was measured for 2 min. *—in control experiments, two concentrations (1 and 200 µmol/L) of AA or DHA were tested.

## Data Availability

The data presented in this study are available on request from the corresponding author.

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
