# Peer review of "Effect of Physiological Concentrations of Vitamin C on the Inhibitation of Hydroxyl Radical Induced Light Emission from Fe2+-EGTA-H2O2 and Fe3+-EGTA-H2O2 Systems In Vitro"

_molecules, 2021, doi:10.3390/molecules26071993_

Round 1
Reviewer 1 Report
- at pH of 7.4 and temperature of 37°C. why at these conditions??. If these conditions are changed, what will happen?.
- I think the authors should support the results with the effect of changing of these conditions depending upon the previous studies (if present).
- EGTA-, please write this abbreviation and the other abbreviations in the 1st writing of it in the manuscript.
- Addition of AA (final concentrations of 1 µmol/L, 5 µmol/L, 10 µmol/l, 25 µmol/L and 50 µmol/L) >>> should be re-written >>>> Addition of AA (final concentrations of 1, 5, 10, 25, and 50 µmol/L)
- Control experiments showed no effect of AA (final concentrations of 1 µmol/L and 200 µmol/L) >>> should be re-written >>>> Control experiments showed no effect of AA (final concentrations of 1, and 200 µmol/L)
- Figure 1 showed the effect of AA to 200 µmol/L. What are the exact used concentrations of AA????
- According to figures, the concentrations od AA are 1 to 200. It is correct??. if correct please clearify it in the manuscript.
- You used DHAA at two concentrations only at 1 and 200. How you can compare with AA althought you did not use the same concentrations?
- I observed that all the figures have strong disruption. Why??. I think the authors can justify this problem
- There are some words with colour highlights in the manuscript such as (within the wavelength range, page 11). Why???
- I think there are no need for the Abbreviations section. The authors can use the complete terms in the 1st description in the manuscript
- there are many typographical errors that should be revised by the authors
Author Response
Responses to comments of the Reviewer 1
1.at pH of 7.4 and temperature of 37°C. why at these conditions??. If these conditions are changed, what will happen?.
2.I think the authors should support the results with the effect of changing of these conditions depending upon the previous studies (if present).
Response (Re) to points 1 and 2: The following statements were added at the end of subsection 3.3. Relevance to human physiology : “Circulating blood has pH of around 7.4 and temperature of 37°. However, locally at the place of inflammation and also in certain solid tumors the tissue environment could be acidic with pH ranging from 5.7 to 7.0 [46]. This may predispose to reduction of Fe3+ to Fe2+ by AA and enhanced generation of •OH radicals. Therefore, pro-oxidant activity of ascorbate cannot be excluded under such circumstances.”
- 3. EGTA-, please write this abbreviation and the other abbreviations in the 1st writing of it in the manuscript.
Re: This is corrected according to the reviewer suggestion.
- 4. Addition of AA (final concentrations of 1 µmol/L, 5 µmol/L, 10 µmol/l, 25 µmol/L and 50 µmol/L) >>> should be re-written >>>> Addition of AA (final concentrations of 1, 5, 10, 25, and 50 µmol/L)
Re: This is corrected throughout the whole manuscript according to the reviewer suggestion.
- 5. Control experiments showed no effect of AA (final concentrations of 1 µmol/L and 200 µmol/L) >>> should be re-written >>>> Control experiments showed no effect of AA (final concentrations of 1, and 200 µmol/L)
Re: This is corrected throughout the whole manuscript according to the reviewer suggestion.
6.Figure 1 showed the effect of AA to 200 µmol/L. What are the exact used concentrations of AA?
- According to figures, the concentrations od AA are 1 to 200. It is correct??. if correct please clearify it in the manuscript.
Re to points 6 and 7: Horizontal axis of Figure 1A shows that we studied 8 concentrations of ascorbic acid (1, 5, 10, 25, 50, 75, 100 and 200 µmol/L). But in control experiments shown in Figure 1B (horizontal axis) only two concentrations were studied: 1 and 200 µmol/L. This is described in the manuscript (Material and Methods, and Results).
- 8. You used DHAA at two concentrations only at 1 and 200. How you can compare with AA althought you did not use the same concentrations?
Re: DHAA was also tested at the same concentrations as AA. Eight concentrations(1, 5, 10, 25, 50, 75, 100 and 200 µmol/L) were used in the case of Fe2+-EGTA–H2O2 and Fe3+-EGTA- H2O2 (horizontal axis of Figures 2A and 4A) and two concentrations (1 and 200 µmol/L) for control experiments (horizontal axis of Figures 2B and 4B). This was also described in the manuscript (Material and Methods, and Results).
- I observed that all the figures have strong disruption. Why??. I think the authors can justify this problem
Re: Sorry, we do not quite understand this question.
10.There are some words with colour highlights in the manuscript such as (within the wavelength range, page 11). Why???
Re: This was the accidental error which is corrected in the revised version of manuscript.
- I think there are no need for the Abbreviations section. The authors can use the complete terms in the 1st description in the manuscript
Re: The “Abbreviations section” was rejected from revised manuscript according to the reviewer suggestion. However, we added the statement “ RLU – relative light units” to figure captions (Figures 1, 2, 3 and 4).
12: there are many typographical errors that should be revised by the authors
Re: This is corrected throughout the whole manuscript
Reviewer 2 Report
The paper entitled "Physiological Concentrations of Vitamin C Inhibit Hydroxyl Radical Dependent Light Emission from Fe2+-EGTA-H2O2 and Fe3+-EGTA-H2O2 Systems In Vitro." presents a interesting study, with a very complete set of analyses.
Furthermore, paper was very well write. My only question is related to the firsts two lines of the results. I could not understand this sentence.
Besides, I think that conclusion must include more objetives conclusions related to health, for example.
Author Response
Responses to comments of the Reviewer 2
1.My only question is related to the firsts two lines of the results. I could not understand this sentence.
Re: The first two sentences of the results were “ UPE of 92.6 µmol/L Fe2+-185.2 µmol/L EGTA- 2.6 mmol/L H2O2 was 2306±910 (2052; 71) RLU (n=11). Incomplete systems Fe2+ - H2O2 and Fe2+-EGTA emitted less light (p<0.05, n=11) 1022±295 (958; 339) RLU and 675±111(678; 137) RLU, respectively.” They were changed in the revised manuscript for; “ The light emission (UPE- ultra weak photon emission) from 92.6 µmol/L Fe2+-185.2 µmol/L EGTA- 2.6 mmol/L H2O2 system was 2306±910 (2052; 71) RLU (n=11). UPE from incomplete control systems Fe2+ - H2O2 and Fe2+-EGTA was significantly lower (p<0.05, n=11) and reached 1022±295 (958; 339) RLU and 675±111(678; 137) RLU, respectively.”
- 2. Besides, I think that conclusion must include more objetives conclusions related to health, for example.
Re: According to reviewer proposal we added at the end of subsection “Conclusions” the following sentence: “ Although, these results were obtained from in vitro experiments they strongly suggest the low risk of pro-oxidant activity of AA in healthy subjects.”
Reviewer 3 Report
Introduction
In the sentence "It should be pointed out that combinations of iron ions with AA and/or H2O2 were widely used for induction of DNA damage in in vitro studies [21-24]." "in vitro" should be written in italic.
Italic should be used for all identical phrases in the whole text.
Discussion
It is quite unusual that we see chemical equations in this section. These should be omitted and explained within the Results section.
Discussion section is lacking the comparisons with other similar studies which should be added.
Also, within the Discussion section authors should elaborate in more details the significance of their finding in the context of disease prevention and treatment and cited appropriate references.
References
The list of references contains large number of very old references that is not good. Authors need to replace old references with the new ones.
Author Response
Responses to comments of the Reviewer 3
- In the sentence "It should be pointed out that combinations of iron ions with AA and/or H2O2 were widely used for induction of DNA damage in in vitro studies [21-24]." "in vitro" should be written in italic.
- Italic should be used for all identical phrases in the whole text.
Re to points 1 and 2: This is corrected throughout the whole revised manuscript
- 3. Discussion. It is quite unusual that we see chemical equations in this section. These should be omitted and explained within the Results section.
- Discussion section is lacking the comparisons with other similar studies which should be added.
Re to points 3 and 4: The description of light emission from Fe2+-EGTA- H2O2 system with its possible mechanism and application for evaluation of anti- or pro-oxidant activity of various compounds was our original discovery. Recently we reported these in two original articles:
- . Nowak, M.; Tryniszewski, W.; Sarniak, A.; Wlodarczyk, A.; Nowak, P.J.; Nowak, D. Light emission from the Fe2+-EGTA-Hâ‚‚Oâ‚‚ system: possible application for the determination of antioxidant activity of plant phenolics. Molecules. 2018, 10, 866.
- Nowak M, Tryniszewski W, Sarniak A, WÅ‚odarczyk A, Nowak PJ, Nowak D. Light emission from the Fe2+-EDTA-ascorbic acid-H2O2 system strongly enhanced by plant phenolic acids. Luminescence. 2019,34,183-192.
Hence, we were not able to compare our results with other similar studies.
In “Discussion section” we described the plausible molecular mechanism of inhibitory effect of ascorbate on light emission from Fe2+-EGTA- H2O2 system. Ascorbic acid can react with H2O2 , superoxide radicals and hydroxyl radials. Moreover, it can effectively reduce Fe3+ to Fe2+ and chelate both these cations. Therefore, to propose the exact mechanism it was necessary to describe all known chemical reactions (chemical equations) leading to hydroxyl radicals generation and light emission from Fe2+-EGTA- H2O2 system. And it was done in the “Discussion section”
- Also, within the Discussion section authors should elaborate in more details the significance of their finding in the context of disease prevention and treatment and cited appropriate references.
Re: It should be pointed out that our results and conclusions were obtained from experiments executed in vitro. Although the conditions of our experiments resembled to some extent those present in human plasma, they are far from real in vivo settings. That is why you have to be careful with transferring our results to preventive medicine and health benefits related to vitamin C ingestion. Therefore we, intentionally avoided the discussion of our results in the context of possible disease prevention.
- The list of references contains large number of very old references that is not good. Authors need to replace old references with the new ones.
Re: The list of cited references in revised manuscript increased by 1 to 49.Twenty six of them were published after 2010 year. On the other hand, the Fenton reaction which leads to generation of hydroxyl radicals was described for the first time in 1876 year. Since that time the chemistry of this process is intensively studied. Therefore, we cited 10 papers published between 1970 and 1999 year.